# Whole Genome Sequence Analysis of a Novel *Apilactobacillus* Species from Giant Honeybee (*Apis dorsata*) Gut Reveals Occurrence of Genetic Elements Coding Prebiotic and Probiotic Traits

**DOI:** 10.3390/microorganisms10050904

**Published:** 2022-04-26

**Authors:** Waqar Ahmad, Shazia Khaliq, Nasrin Akhtar, Jamilah El Arab, Kalsoom Akhtar, Satya Prakash, Munir A. Anwar, Nayla Munawar

**Affiliations:** 1Industrial Biotechnology Division, National Institute for Biotechnology and Genetic Engineering College, Pakistan Institute of Engineering and Applied Sciences (NIBGE-C, PIEAS), Faisalabad 38000, Pakistan; waqarahmadhu424@gmail.com (W.A.); skhaliq1976@gmail.com (S.K.); nasrin_379@yahoo.com (N.A.); kalsoom1967@gmail.com (K.A.); 2Biomedical Technology and Cell Therapy Research Laboratory, Department of Biomedical Engineering, Faculty of Medicine, McGill University, 3775 University Street, Montreal, QC H3A 2B4, Canada; satya.prakash@mcgill.ca; 3Department of Chemistry, College of Sciences, United Arab Emirates University, Al-Ain 15551, United Arab Emirates; 202170109@uaeu.ac.ae

**Keywords:** giant honeybee (*Apis dorsata*), gut, *Apilactobacillus*, probiotics, prebiotics, glucans, dextran, glucosyltransferase, exopolysaccharides

## Abstract

*Apilactobacillus* spp. are classified as obligate fructophilic lactic acid bacteria (FLAB) that inhabit fructose-rich niches such as honeybee gut. Lactic acid bacteria are an important component of the gut microbiome and play a crucial role in maintaining gut health. In this study, a new FLAB strain HBW1, capable of producing glucan-type exopolysaccharide, was isolated from giant honeybee (*Apis dorsata*) gut and subjected to whole genome sequencing (WHS) to determine its health-beneficial traits. The genome size of the isolate was 1.49 Mb with a GC content of 37.2%. For species level identity, 16S rDNA sequence similarity, genome to genome distance calculator (dDDH), and average nucleotide identity (ANI) values were calculated. Phylogenetic analysis showed that the isolate HBW1 belongs to the *Apilactobacillus* genus. The dDDH and ANI values in comparison with closely clustered *Apilactobacillus kunkeei* species were 52% and 93.10%, respectively. Based on these values, we concluded that HBW1 is a novel species of *Apilactobacillus,* and we propose the name *Apilactobacillus waqarii* HBW1 for it. Further, WHS data mining of HBW1 revealed that it harbors two glucosyltransferase genes for prebiotic glucan-type exopolysaccharide synthesis. Moreover, chaperon (*clp*) and methionine sulfoxide reductase (*msrA*, *msrB,* and *msrC*) genes as well as nutritional marker genes for folic acid (*folD*) and riboflavin biosynthesis (rib operon), important for conferring probiotic properties, were also detected. Occurrence of these genetic traits make HBW1 an excellent candidate for application to improve gut function.

## 1. Introduction

Studies during the last few years have shown that the gut and brain are interconnected both physically and chemically through a communication network called the gut–brain axis [1,2,3]. This implies that the brain and gut may reciprocally affect each other’s health. Diet and gut microbes play an important role in maintaining the gut health. Diet is one of the major factors that modulates the composition of gut microbiota [4] thus affecting brain health. Therefore, modulating the gut microbiome may help in improving brain health. Certain diet ingredients, such as probiotics and prebiotics, have the potential to play a crucial role in this respect. Indeed, recently reviewed research data have shown that these components are helpful in treatment of brain-related and other disorders [5,6,7].

Probiotics are defined as “live microorganisms that, when administered in adequate amounts, confer a health benefit to the host” [8]. These include mostly certain lactic acid bacteria (LAB) and bifidobacteria [9]. A prebiotic has been defined as “a substrate that is selectively utilized by host microorganisms conferring a health benefit” [10]. Prebiotics are fibrous substrates that resist enzymatic activity in the upper part of human gastrointestinal tract; thus, they are not digested in the small intestine. Consequently, they reach the colon unmetabolized and are fermented by the intestinal microbiota. This process positively regulates the activity of the specific intestinal microbiome that beneficially interacts with the host by the producing short chain fatty acids (SCFAs) and vitamins [11,12]. Chemically, prebiotics are oligo/polymers of carbohydrates such as fructooligosaccharides (FOS), galactooligosaccharides (GOS), xylooligosaccharides (XOS), etc., [12,13].

In addition to their probiotic effects, LAB are also of particular interest for the production of exopolysaccharides (EPSs) that have potential prebiotic functionality. For this purpose, LAB produce sucrase enzymes, which synthesize glucose or fructose polysaccharides called as glucans and fructans, respectively, using sucrose as a substrate [14,15]. The enzymes involved in glucan synthesis are known as glucansucrases and belong to the glycoside hydrolase family 70 (GH70) at carbohydrate-active enzymes database (http://www.cazy.org) (accessed on 10 March 2022) [16]. Depending on the type of linkages they introduce between glucose residues in the polysaccharide, glucansucrases are categorized as dextransucrases—synthesizing dextran (containing mainly α-(1-6) linkages); mutansucrases—synthesizing mutan (α-(1-3 linkages); reuteransucrases—synthesizing reuteran (α-(1-4) linkages); and alternansucrases that synthesize alternan (having alternating α-(1-6) and α-(1-3)-linkages) [15,17,18].

Fructophilic lactic acid bacteria (FLAB) is a specific subgroup of LAB that has been described and characterized recently [19]. FLAB have gained considerable attention due to their potential human health beneficial effects and for their proximity to lactic acid bacteria that are accepted as safe [20]. They live in symbiosis with insects such as honeybees that have special diets [21,22]. FLAB preferably use fructose as a substrate for their growth and inhabit only those niches that are rich in fructose content. They are also found in different fruit and vegetable food matrices such as banana, grapes, figs, durian fruit, legumes, and cocoa beans [23].

Among FLAB, there are some other members of the genus *Lactobacillus* that have recently been reclassified as *Apilactobacillus* to underline their specific adaptation to bees [24]. The genus *Apilactobacillus* includes different bacterial species; among them, *Apilactobacillus kunkeei* (basonym *Lactobacillus kunkeei*) and *Apilactobacillus apinorum* (basonym *Lactobacillus apinorum*) have adapted to bees [25]. These lactic acid bacteria constitute an important component of human gut microbiota as well and could play crucial role in maintaining gut health due to their prebiotic synthesizing potential or probiotics traits. In the present study, a new *Apilactobacillus* spp. strain, designated as HBW1, was isolated from giant honeybee (*Apis dorsata*) gut and subjected to whole genome sequencing with an aim to identify its prebiotic EPS synthesizing gene(s) and the genetic elements important for conferring probiotic properties to the isolate. We have proposed the name *Apilactobacillus waqarii* HBW1 for this isolate.

## 2. Materials and Methods

### 2.1. Culturing of Bacteria

A wild type giant honeybee (*Apis dorsata*), found in Southeast Asia, was sampled from a botanical garden in Faisalabad region of Pakistan. The outer surface of the bee was sterilized with 70% ethanol, and its gut was removed under sterile condition. The gut was transferred to sterilized De Man-Rogosa-Sharpe (MRS)-sucrose medium having the composition described previously [26] and incubated at 30 °C for 24 h. After visual appearance of growth, serial dilutions of the cultured broth were made and streaked on agar plates of the same medium to obtain single colonies. A single colony having a slimy mucoid appearance, indicating the production of EPS, was selected and transferred to sterilized fresh MRS-sucrose medium for growth. The process of transferring to solid and liquid media was repeated twice further to assure the isolation of pure culture, which was designated as HBW1.

### 2.2. Exopolysaccharide Production by the Isolate

To confirm the EPS production HBW1, MRS medium, supplemented with sucrose (20% *w*/*v*), was inoculated with HBW1 and incubated for 48 h in shaking incubator. To check for EPS production, a small aliquot of culture was run on thin layer chromatography (TLC) plate (Silica gel 60 F254; Merck, Darmstadt, Germany) for 6 h, after which it was air dried, and the sugar spots were visualized by developing with a solution containing 5% sulfuric acid in methanol, as described previously [27].

### 2.3. Genome Sequencing

The Genomic DNA of the isolate was extracted using a Thermo Scientific Gene JET Genomic DNA extraction kit (#K0721). The whole genome sequencing was carried out commercially by MicrobesNG (Birmingham, United Kingdom) using Illumina next generation sequencing at minimum coverage of 30×. The whole-genome sequencing data and bio-project/bio-sample information of the *Apilactobacillus* species (HBW1) reported in the present study are available at NCBI database (Accession numbers given in Table 1).

### 2.4. Genome Analysis

The genome annotation was carried out using NCBI Prokaryotic Genome Annotation Pipeline (PGAP) and Rapid Annotation using Subsystem Technology (RAST) server version 2.0 (Classic RAST default settings) [28]. The genomic digital DNA Hybridization (dDDH) values were calculated using the Genome-to-Genome Distance Calculator (GGDC) DSMZ (https://www.dsmz.de/services/online-tools/genome-to-genome-distance-calculator-ggdc) (accessed on 12 April 2022) [29]. Average Nucleotide Identity (ANI) values were determined using the Kostas Lab server (http://enve-omics.ce.gatech.edu/ani/) (accessed on 12 April 2022) [30] with default parameters.

### 2.5. Taxonomic Evaluation

For taxonomic studies, 16S rRNA gene sequences of the closely related species were taken from EzBioCloud and the National Center for Biotechnology Information (NCBI) databases [31]. The phylogenetic analysis and trees were constructed on the basis of whole genome sequences using the Type (Strain) Genome Server (TYGS) [29]. Trees were inferred with FastME 2.1.6.1 [32] from the Genome BLAST Distance Phylogeny (GBDP) distances calculated from genome sequences. The branch lengths were scaled in terms of GBDP distance formula d5. The numbers above branches were GBDP pseudo-bootstrap support values >60% from 100 replications, with an average branch support of 89.9 and 71.4% for 16S rRNA and genome sequences, respectively. The trees were rooted at the midpoint [33].

### 2.6. EPS Producing Genes

The genome sequence of HBW1 was also analyzed for the presence of genes specific for EPS synthesis and probiotic traits. Moreover, the EPS synthesizing gene was compared with top related strains obtained from EzBioCloud [31]. The phylogenetic tree was constructed using MEGA7 software by the neighbor joining method.

### 2.7. Submission Information

The whole genome of HBW1 was submitted to the NCBI database. The details are in Table 1.

## 3. Results and Discussion

### 3.1. EPS Production

Thin layer chromatographic analysis showed that the isolate *Apilactobacillus waqarii* HBW1 produced glucan type EPS using sucrose as a substrate (Figure 1). Previously, the presence of glucansucrase enzymes responsible for glucan type EPS synthesis has only been reported in *Apilactobacillus kunkeei* species among apilactobacilli [14]. Therefore, *A. waqarii* is only the second species among *Apilactobacillus* genus that produces glucan type EPS. 

### 3.2. Genome Sequence and Annotation

The genome assembly was classified as undecided for the potential contaminations as determined by ContEst16S (Contamination Estimator by 16S) algorithm. As per NCBI PGAP analysis, the draft genome sequence of the isolate HBW1 comprised a total length of (1.49 Mb); contig count (38); N50 (229,491 bp), L50 (3), and G + C content (37.2%). Moreover, a total of 1340 protein-coding sequences and 75 RNAs were found in the genome (Table 2). Further ClassicRAST based functional gene subsystem clustering analysis revealed that 247 subsystems were present in the genomic island of the HBW1 (Table 2). The subsystems representing the amino acids and derivatives (79 ORFs); cofactors, vitamins, prosthetic groups, pigments (69 ORFs); carbohydrate metabolism (70 ORFs); protein metabolism (194 ORFs); fatty acids, lipids, and isoprenoids (56 ORFs) were present in the genome of HBW1. Furthermore, the subsystems connected with membrane transport (31 ORFs), stress response (33 ORFs), and sulfur metabolism (3 ORFs) were also identified (Figure 2).

### 3.3. Phylogenetic and Genome Based Classification at Species Level

Taxonomic evaluation of the isolate HBW1 was carried out by systematically using a combination of 16S rRNA gene similarity and Overall Genome-Related Index (OGRI) that included ANI and dDDH. In this regard, the first step was to determine the strains that were closely related to HBW1. For this purpose, the 16S rRNA gene sequence of HBW1 was submitted to the NCBI (for BLASTn search) and EzBioCloud databases [31,34]. The results of the NCBI BLASTn and EzBioCloud showed a sequence similarity of 16S rRNA gene of HBW1 with other congener species of the genus *Apilactobacillus* (*Kunkeei*, *Apinorum*, *Bombintestini*, *Timberlakei*, *Micheneri*, *Quenuiae,* and *Ozensis*)—maximum similarity (100%) of the HBW1 was found with *Apilactobacillus Kunkeei* DSM 12,361 YH-15^T^ [JXDB01000004], followed by *Apilactobacillus apinorum* Fhon13N^T^ [JX099541], 98.82% (Table 3). However, genome-based phylogenetic analysis of the closely related whole genome sequences showed that the HBW1 formed a separate branch from *Apilactobacillus kunkeei* species (Figure 3). Further, to establish a more specific taxonomic position at the species level, a comparison of the genome of the HBW1 was carried out to its closely-related type strains using ANI and dDDH values (Table 3). According to the current bacterial taxonomy, the projected and generally accepted dDDH and ANI values are 70% and 95–96%, respectively, between genomes of the same species [32]. A comparison of the HBW1 and the close neighbor *Apilactobacillus kunkeei* DSM 12,361 strain YH15^T^ revealed dDDH value of 52.0 and the ANI value 93.10 supporting HBW1 as a potential new species (Table 3). Taking together the 16S rRNA gene, dDDH, and ANI, the isolate HBW1 described here is a new species of *Apilactobacillus* genus, and we propose the name *Apilactobacillus waqarii* HBW1 for it. Previous studies showed that geographical location and even developmental stages can influence the diversity in composition of honeybee gut microbiota [35,36,37]. Among other factors, exposure to synthetic chemicals such as pesticides also determine the type of microbiota inhabiting honeybee gut [38]. Owing to distinct geographical and environmental conditions at the habitat of the host giant honeybee *A. dorsata*, some or all of these factors could have contributed to the occurrence of this novel *Apilactobacillus* species. 

### 3.4. Genetic Traits of the Isolate HBW1 Important for Gut Function 

#### 3.4.1. Putative Genes for EPS Synthesis

Mining of the whole genome sequencing data of HBW1 revealed that it harbored two genes coding for putative glucosyltransterase proteins, designated as GTF1-HBW1 and GTF2-HBW1, responsible for EPS synthesis. In the phylogenetic tree, the GTF1-HBW1 and GTF2-HBW1 clustered with glucosyltransferase of closely related species of *Apilactobacillus kunkeei* (WP-220382206.1) and dextransucrase of *Apilactobacillus kunkeei* (KPN80157.1), respectively (Figure 4). Interestingly, the GTF1-HBW1 and GTF2-HBW1 fell in highly divergent clades depicting that these proteins would be responsible for very different types of glucans, one of which is likely to be a dextran.

Several species of *Apilactobacillus* (basonym *Lactobacillus*) genus have been isolated from honeybee *Apis mellifera* gut [39]. Studies based on metagenomic analysis have shown that these FLAB are also widely distributed among honeybee genera *A. mellifera* [40] (Nowak et al. 2021), *A. florea* [36], and *Apis dorsata* [35]. However, EPS synthesizing enzymes have only been characterized from *Apilactobacillus kunkeei* [41,42]. In contrast, among LAB, production of dextran type EPS is commonly known in *Leuconostoc* and *Weissella* species [14,43,44]. Similar to *A. waqarii* HBW1, some of them were also reported to have multiple genes responsible for the production of glucan-type of EPSs [45]. 

In addition to some fructans and fructooligosaccharides (FOS), the prebiotic potential of dextran has also been documented. Using a batch-culture fermentation system designed to simulate transit through the large intestine, dextran has been demonstrated to elicit a bifidogenic effect similar to the well-known prebiotic fructooligosaccharides (FOS), which also resulted in decreased levels of undesirable bacteria such as clostridia and bacteroides [46]. Similarly, some linear and branched dextrans have been found to increase *Bifidobacterium* populations significantly during fermentation by human fecal microbiota [47].

#### 3.4.2. Genetic Elements Conferring Potential Probiotic Characteristics

Mining of the whole genome sequence data of *A. waqarii* HBW1 revealed that it also harbored certain genes that are important for conferring probiotic properties to bacteria exerting beneficial effects on host health. Genes that are responsible for the active removal of stressors including DNA and the protein protection and repair system, e.g., *clpATPase* (chaperon) reported to have bile salt hydrolase and acid tolerance activity [48] were detected in the genome of HBW1 as shown in Table 4. The location of these genes on the HBW1 genome is shown in Figure 5. The CLP chaperone protein is a mimetic of the anorexigenic α-melanocyte stimulating hormone (α-MSH). Several recent studies on mice model have shown that the prevalence of this gene in enterobacteria resulted in reduced adiposity and weight gain in obesity rodent models [49,50,51]. This fact is further strengthened by the finding that low enterobacterial Clp B gene abundance was observed in the microbiota of obese humans [52]. Other probiotic genes such as methionine sulfoxide reductase genes *msrA*, *msrB*, and *msrC* are also very important, because reactive oxygen species (ROS) oxidize the methionine residues in proteins, resulting in the production of methionine-S-sulfoxides [Met-S-(O)] and methionine-R-sulfoxides [Met-R-(O]. These oxidized methionine residues can be repaired by the antioxidant enzymes, Met-S-(O) reductase (*MsrA*) and Met-R-(O) reductase (*MsrB*) [53]. We have also identified the nutritional marker gene genes for folic acid (*folD*) and riboflavin biosynthesis (rib operon). These nutritional marker genes have the potential ability to synthesize and transport B vitamins-riboflavin [54]. Synthesis of B vitamins is a desirable trait of probiotic bacteria since the human body does not synthesize these vitamins [55].

## 4. Conclusions

A novel species of fructophilic lactic acid bacteria *Apilactobacillus*, with the proposed name *Apilactobacillus waqarii* HBW1, was isolated from giant honeybee (*Apis dorsata*) gut. The genome of HBW1 harbors two EPS synthesis genes and genetic elements important for conferring probiotic properties, making it an excellent candidate for application to improve gut function. In future studies, the composition and glcosydic linkage analysis of the purified EPS products synthesized by heterologously expressed GTF proteins of HBW1 would resolve the exact chemical structure of these putative glucans. 

## Figures and Tables

**Figure 1 microorganisms-10-00904-f001:**
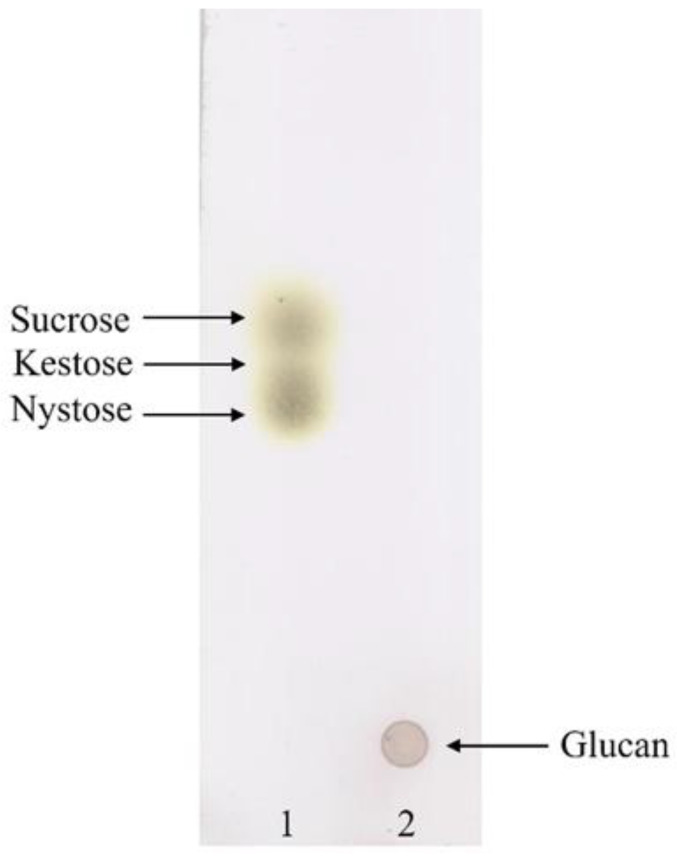
Thin layer chromatographic analysis for glucan synthesis by *A. waqarii* HBW1. 1: Standard; 2: Sample of HBW1 culture grown on MRS-sucrose medium.

**Figure 2 microorganisms-10-00904-f002:**
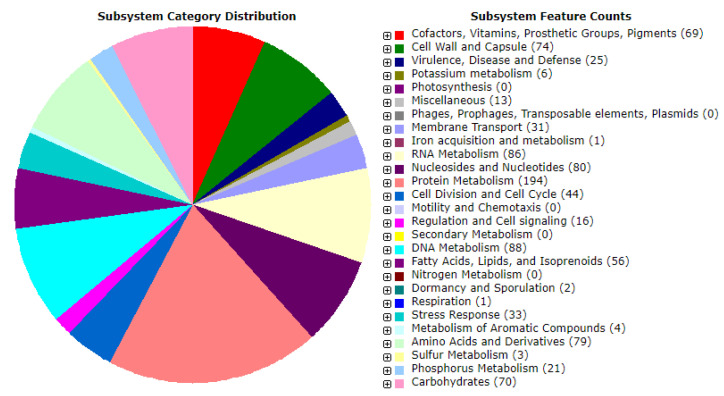
Genome annotation of *Apilactobacillus waqarii* HBW1 using the ClassicRAST online annotation server showing different types of subsystems.

**Figure 3 microorganisms-10-00904-f003:**
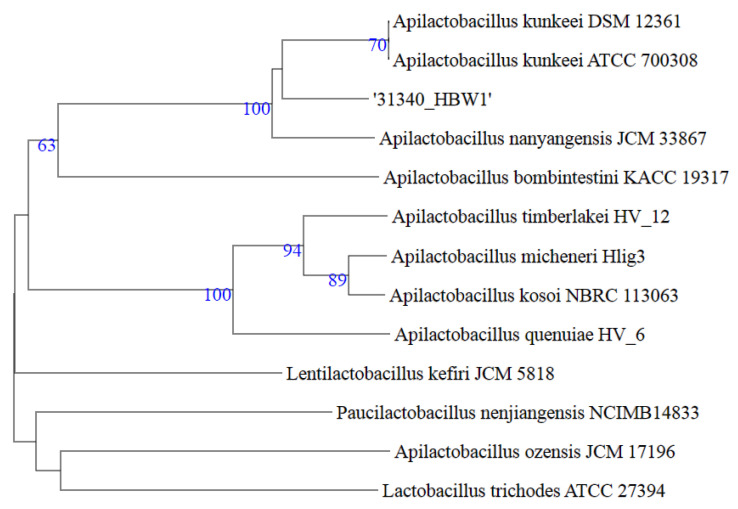
Whole genome based phylogenetic tree of *Aplilactobacillus waqarii* HBW1 generated by species tree generated by Type (Strain) Genome Server (TYGS). The numbers above branches are GBDP pseudo-bootstrap support values > 60% from 100 replications.

**Figure 4 microorganisms-10-00904-f004:**
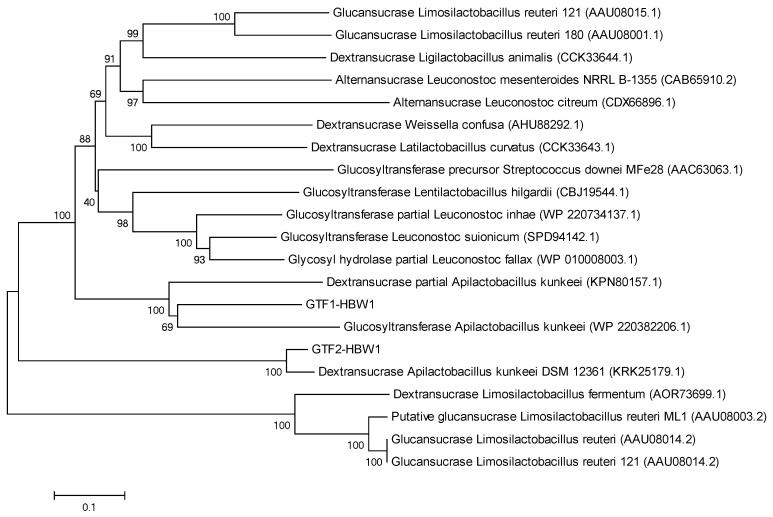
Phylogenetic tree showing the relationship of two glycosyltransferase genes, i.e., GTF1-HBW1 and GTF2-HBW1 of the isolate *Apilactobacillus waqarii* HBW1 with related genes from other bacteria. The tree was constructed with MEGA7 using the neighbor joining method.

**Figure 5 microorganisms-10-00904-f005:**
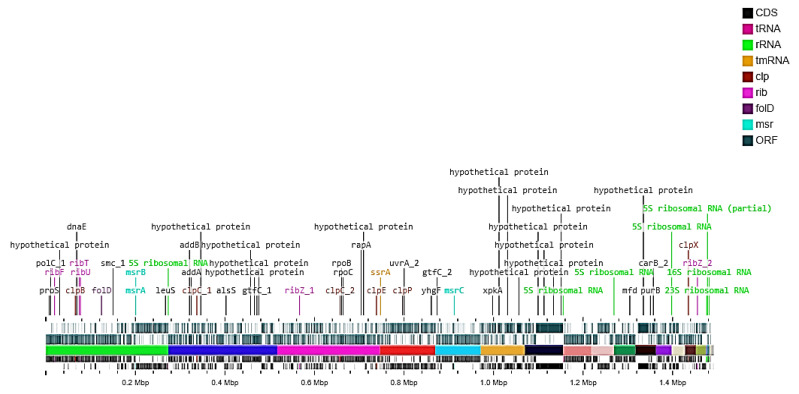
Linear map of genomic DNA of *Apilactobacillus waqarii* HBW1showing the position of probiotic traits of related genes in different colors generated through the online CGview server.

**Table 1 microorganisms-10-00904-t001:** NCBI Submission details of HBW1 Whole Genome Sequencing Data.

Description	Information
Submission ID	SUB9890030
Bio project ID	PRJNA740110
Bio Sample	SAMN19820174
Accession No	JAHQYH000000000
Organism	*Apilactobacillus* sp. HBW1

**Table 2 microorganisms-10-00904-t002:** General genomic attributes of *Apilactobacillus waqarii* strain HBW1 predicted by the NCBI genome annotation pipeline.

Feature	Value
Genome size	1.49 Mb
Genes (total)	1419
G + C content	37.2%
N50	229,491
L50	3
Number of contigs	38
CDSs (total)	1344
Genes (coding)	1340
CDSs (with protein)	1340
Genes (RNA)	75
rRNAs	5, 4, 1 (5S, 16S, 23S)
Complete rRNAs	4, 1, 1 (5S, 16S, 23S)
Partial rRNAs	1, 3 (5S, 16S)
tRNAs	62
ncRNAs	3
Pseudo genes (total)	4
CDSs (without protein)	4
Pseudo genes (ambiguous residues)	0 of 4
Pseudo genes (frameshifted)	1 of 4
Pseudo genes (incomplete)	1 of 4
Pseudo genes (internal stop)	2 of 4
Subsystems	247
Carbohydrate metabolism	70

**Table 3 microorganisms-10-00904-t003:** Taxonomic evaluation of HBW1 on the basis of comparison of its 16S rRNA gene sequence similarity and overall genome-related index values with related type strains.

Strain Name and Accession No.	16S% Identity	dDDH (%)	ANI (%)
*Apilactobacillus kunkeei* DSM 12,361 YH15T	100.00	52.00	93.10
[JXDB01000001]			
*Apilactobacillus apinorum* Fhon13N	98.82	24.50	82.44
[KQ440395]			
*Apilactobacillus bombintestini* BHWM-4	97.67	22.70	78.18
[NZCP03262.1]			
*Apilactobacillus timberlakei* HV-12	96.86	19.80	74.67
[QUAP00000000.1]			
*Apilactobacillus micheneri* Hlig3	95.45	NA	NA
[KT833121]			
*Apilactobacillus ozensis* JCM17196	94.38	22.00	65.38
[AYYQ00000000.1]			

NA: Genome sequence is not available.

**Table 4 microorganisms-10-00904-t004:** Probiotic related genes present in *Apilactobacillus waqarii* HBW1 genome detected by the RAST annotation server.

Genes	Length	Strand	Putative Function	Response
**DNA and protein protection and repair clpATPase (chaperon)**				Acid and bile tolerance
*clpB*	2583 bp	+	ATP-binding subunit *clpB*	
*clpC*	2097 bp	+	ATP-binding subunit *clpC*	
*clPE*	2172 bp	−	ATP-binding subunit *clpE*	
*clpP*	594 bp	+	ATP-binding subunit *clpP*	
*clpX*	1236 bp	+	ATP-binding subunit *clpX*	
**Methionine sulfoxide reductase**				Persistence capacity in vivo
*msrA*	522 bp	−	Methionine sulfoxide reductase A	
*msrB*	483 bp	−	Methionine sulfoxide reductase A	
*msrC*	465 bp	−	Methionine sulfoxide reductase A	
**Folic acid biosynthesis**				Folic acid biosynthesis
*folD*	858 bp	+	Methylenetetrahydrofolate dehydrogenase	
**Riboflavin biosynthesis operon**				Riboflavin biosynthesis
*Ribf*	939 bp	+	Riboflavin kinase/FMN adenylyltransferase	
*Ribu*	582 bp	+	Riboflavin transporter	
*Ribt*	369 bp	+	Riboflavin biosynthesis *RibT* protein	
*Ribz1*	1407 bp	−		
*Ribz2*	1446 bp	−		

## Data Availability

The whole genome data presented in this study are openly available in NCBI database at https://www.ncbi.nlm.nih.gov/search/all/?term=JAHQYH000000000 (accessed on 23 June 2021) under the accession number JAHQYH000000000.

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
