# Peer review of "Whole Genome Sequence Analysis of a Novel Apilactobacillus Species from Giant Honeybee (Apis dorsata) Gut Reveals Occurrence of Genetic Elements Coding Prebiotic and Probiotic Traits"

_microorganisms, 2022, doi:10.3390/microorganisms10050904_

Round 1
Reviewer 1 Report
In this study, an EPS producing bacterial HBW1 was isolated from wild honeybee gut in Pakistan and then subjected to whole genome sequencing. Results revealed that HBW1 could be a novel specie of Apilactobacillus genus. And they found two EPS synthesis genes and certain genetic elements important for conferring probiotic properties.
As it is, this is an interesting study having potential to improve gut health of honey bees.
It will be important to clearly define the sampling protocols and distribution of this n.s. bacteria. On what species of honey bee under what condition and if this could be prevailing in some area etc.
Authors had compared 16S rRNA sequence to closely related species. However it is not still clear how and where this potential new species can be located (separated) from the existing. Better provide the phylogenic relationship among the congeners.
Apilactobacillus (Kunkeei, apinorum, bombintestini, timberlakei, micheneri, quenuiae and ozensis).==> Apilactobacillus (kunkeei, apinorum, bombintestini, timberlakei, micheneri, quenuiae and ozensis).
Author Response
Reviewer 1
Comment: It will be important to clearly define the sampling protocols and distribution of this n.s. bacteria. On what species of honeybee under what condition and if this could be prevailing in some area etc.
Response: Thank you for this suggestion. Please see the modified Materials and Methods section 2.1. (Lines: 100-114)
Comment: Authors had compared 16S rRNA sequence to closely related species. However, it is not still clear how and where this potential new species can be located (separated) from the existing. Better provide the phylogenic relationship among the congeners. (Apilactobacillus (Kunkeei, apinorum, bombintestini, timberlakei, micheneri, quenuiae and ozensis).
Response: As per suggestion of the worthy reviewer, we have added text (Section 3.3, Lines: 239-247), made new entries in Table 3, and added a new figure (Figure 3) to show the phylogenetic relation of HBW1 with the congeners.
Reviewer 2 Report
See comments in a pdf file.

Author Response
Reviewer 2
Comment: Line 22: put “spp.” after Apilactobacillus.
Response: “spp” has been added after Apilactobacillus as per suggestion of the worthy reviewer.
Comment: The Abstract is written with a high level of generality in terms of the results; the results should be better specified.
Response: The results have been more specified now, as suggested by the reviewer. (Please see the Abstract)
Comment: Lines 49-50: give references to the definition; put the latest definition of probiotics and prebiotics, see ISAPP; see: https://pubmed.ncbi.nlm.nih.gov/24912386/
Response: As per suggestion of the worthy reviewer, the latest definitions of Probiotics and Prebiotics, and the suggested reference have been added. Please see new Lines: 52-56 and references 8,10.
Comment: Authors should use the term “microbiota” or “microbiome” instead of incorrect “microflora” in relation to gastrointestinal tract, as no plants are present in the system.
Response: The suggested change has been incorporated.
Comment: Lines 39-71: this text is irrelevant to the research topic and article. It should apply to honeybees, not humans or generally.
Response. The point raised by the worthy reviewer is also important as HBW1 was isolated from the honeybee gut and could have health beneficial effects on honeybees, but the present manuscript has been written in the context of prebiotic and probiotic potential of the isolate with reference to human health. Please see the sections 3.4.1 and 3.4.2 for details.
Comment: Lines 85-91: give detailed aims of the research. In this place no results and no conclusions should be given.
Response. The suggested change has been incorporated in the manuscript (new Line # 91-97 )
Comment: Line 94: specie? Correct “specie” to “species”. English language needs to be spell-checked.
Response: The suggested correction has been made throughout the manuscript.
Comment: Paragraph 2.1 title: culture of what? It should be divided into 2 points: culturing of LAB and EPS production. Give more detailed description.
Response: The changes suggested by the worthy reviewer have been incorporated; a detailed description has been given; section 2.1 is divided into two sections and the subsequent section numbers updated accordingly
Comment: Why the new species name is “waqarii”?
Response. The stain was isolated by the first author “Waqar Ahmad”, we suggested the name of the isolate on his name, similar to Lactobacillus johnsonii, Lactobacillus gasseri etc.
Comment: All abbreviations should be defined, when used for the first time.
Response: All the abbreviations have been cross-checked and defined as per the direction of the worthy reviewer.
Round 2
Reviewer 1 Report
This study revealed a novel species of Apilactobacillus showed EPS element producing genetic codes.
In the title, honey bee should replaced by "giant honey bee (Apis dorsata). The same should be applied in all text and discussed the possible logistic reason of having this novel species.
And if the LAB and EPS are common for the genus Apilactobacillus also have to be cleared.
Authors had mentioned simply that giant honey bees were collected and gut bacteria were incubated to find this bacteria.
However, the prevalence of this bacteria in Apis dorsata or in other species of honey bees were not studied. Also if the new bacteria can be commonly sampled from Apis dorsata is still in question. For this, authors could provide some information even with the form of personal communication or observation which has relatively lower confidence but could provide some cues.
Table 4. Text arrangement could save space
Fig. 5. If the figure arranged vertically rather than horizontally, information authors would like to convey could be clear.
Reviewer 2 Report
I have no more comments.
Author Response
The worthy reviewer have no more comments